# Detection and Analysis of VOCs in Cherry Tomato Based on GC-MS and GC×GC-TOF MS Techniques

**DOI:** 10.3390/foods13081279

**Published:** 2024-04-22

**Authors:** Sihui Guan, Chenxu Liu, Zhuping Yao, Hongjian Wan, Meiying Ruan, Rongqing Wang, Qingjing Ye, Zhimiao Li, Guozhi Zhou, Yuan Cheng

**Affiliations:** 1Vegetable Research Institute, Zhejiang Academy of Agricultural Sciences, Hangzhou 310021, China; guansihui107@126.com (S.G.); liuchenxu@zaas.ac.cn (C.L.); yaozp@zaas.ac.cn (Z.Y.); hjwan@zaas.ac.cn (H.W.); ruanmy@zaas.ac.cn (M.R.); wangrq@zaas.ac.cn (R.W.); yeqj@zaas.ac.cn (Q.Y.); zhimiaoli@zaas.ac.cn (Z.L.); zhougz@zaas.ac.cn (G.Z.); 2College of Agriculture, Shihezi University, Shihezi 832003, China

**Keywords:** cherry tomato, VOCs, HS-SPME-GC-MS, HS-SPME-GC×GC-TOFMS, rOAV

## Abstract

Volatile organic compounds (VOCs) play a significant role in influencing the flavor quality of cherry tomatoes (*Solanum lycopersicum* var. cerasiforme). The scarcity of systematic analysis of VOCs in cherry tomatoes can be attributed to the constraints imposed by detection technology and other contributing factors. In this study, the cherry tomato cultivar var. ‘Zheyingfen1’ was chosen due to its abundant fruit flavor. Two detection technology platforms, namely the commonly employed headspace solid-phase microextraction—gas chromatography–mass spectrometry (HS-SPME-GC-MS) and the most advanced headspace solid-phase microextraction—full two-dimensional gas chromatography–time-of-flight mass spectrometry (HS-SPME-GC×GC-TOFMS), were employed in the analysis. The VOCs of cherry tomato cultivar var. ‘Zheyingfen1’ fruits at red ripening stage were detected. A combined total of 1544 VOCs were detected using the two aforementioned techniques. Specifically, 663 VOCs were identified by through the HS-SPME-GC-MS method, 1026 VOCs were identified by through the HS-SPME-GC×GC-TOFMS, and 145 VOCs were identified by both techniques. The identification of β-ionone and (E)-2-nonenal as the principal VOCs was substantiated through the application of the relative odor activity value (rOAV) calculation and subsequent analysis. Based on the varying contribution rates of rOAV, the analysis of sensory flavor characteristics revealed that cherry tomato cultivar var. ‘Zheyingfen1’ predominantly exhibited green and fatty attributes, accompanied by elements of fresh and floral flavor characteristics. In conclusion, our study conducted a comprehensive comparison of the disparities between these two methodologies in detecting VOCs in cherry tomato fruits. Additionally, we systematically analyzed the VOC composition and sensory flavor attributes of the cherry tomato cultivar var. ‘Zheyingfen1’. This research serves as a significant point of reference for investigating the regulatory mechanisms underlying the development of volatile flavor quality in cherry tomatoes.

## 1. Introduction

*Solanum lycopersicum* L., commonly known as tomato, is a highly cultivated vegetable crop globally, renowned for its abundant content of lycopene, vitamin C, minerals, and various other essential nutrients, thereby possessing significant economic significance [1,2]. *Solanum lycopersicum* var. cerasiforme, a cultivar within the *Solanum* genus of the Solanaceae family, commonly referred to as mini-tomato, possesses notable attributes such as vibrant pigmentation, a delightful blend of sweetness and tartness, and a high nutritional content. This fresh tomato variant effectively addresses the market demand for off-season fruit, thereby garnering considerable consumer appreciation [3]. The cherry tomato cultivar var. ‘Zheyingfen1’ represents China’s inaugural domestically cultivated single-sex cherry tomato variety, distinguished by its elevated sugar content and amino acid composition. This fruit exhibits a robust aroma, while surpassing ordinary cultivars in terms of flavor and overall quality.

The flavor of any fruit is the sum of interactions between taste and olfaction. For the tomato, sugars and acids activate taste receptors, while a diverse set of volatile compounds activate olfactory receptors. Volatiles, in particular, are essential for good flavor. The primary factor influencing this phenomenon is the interplay between VOCs, including hexal, cis-3-hexenal, and β-ionone [4]. Huang Sanwen’s team and collaborators conducted a thorough sensory evaluation of 160 tomato samples sourced from 101 distinct germplasms. This evaluation led to the identification of 33 significant flavor substances in tomatoes that influence consumers’ preferences. These substances encompass glucose, fructose, citric acid, and malic acid, as well as 29 VOCs. This study unveiled a comprehensive study of the chemistry and genetics of tomato flavor [5,6]. The investigation into VOCs in tomato fruit commenced in the 1950s, and as research progressed, an increasing number of VOCs were discovered [7]. Lee et al. [8] employed HS-SPME-GC-MS to investigate the variations in VOCs among eight distinct tomato varieties cultivated in both greenhouse and open field environments. A total of 40 VOCs were identified. Notably, the concentrations of hexal, para-cymene, and (E)-2-hexenal in the ‘TAMU’ tomato variety exhibited significant disparities compared to the other tomato varieties. The odor characteristics of volatiles with different functional groups greatly differ. Alcohols, for example, are responsible for the sweetness of tomatoes and play a crucial role in augmenting their overall flavor profile [9]. Aldehydes, characterized by their green plant scent reminiscent of freshly harvested grass or leaves, have been demonstrated to heighten the freshness of tomatoes [10]. Ketones, with their floral, fruity, and sweet characteristics, are highly valued for enriching the aroma profile of tomatoes [11]. Phenols emit a potent odor that may elicit discomfort in individuals.

It is noteworthy that the research on the formation and regulation of crucial compounds, such as sugar and acid, which play a significant role in determining the flavor of tomatoes, has advanced considerably. However, in contrast, the investigation into the substances responsible for determining the aroma of tomatoes is still in its nascent stages. In recent years, the advancements in HS-SPME-GC-MS detection technologies and platforms have been instrumental [12,13]. The systematic examination of VOC constituents in tomato fruits has garnered significant interest among researchers in the respective fields [14,15,16,17]. Contemporary advancements in flavor sensory analysis technology enhance conventional analysis techniques by incorporating not only subjective sensory evaluation but also analytical instruments and intelligent sensory instruments as aids for sensory evaluation, thereby enhancing the determinism and accuracy of sensory analysis [18]. The HS-SPME technique, which eliminates the need for solvents and offers simplicity, ease of operation, and rapidity, renders it particularly well suited for the extraction of volatile components. When coupled with GC-MS, HS-SPME emerges as a prevalent technology for the detection and analysis of VOCs in fruits and vegetables, including tomatoes [19,20]. The HS-SPME-GC-MS technique exhibits notable sensitivity when analyzing high-molecular-weight substances, yet it presents specific constraints when attempting to identify low-molecular-weight compounds and trace substances, particularly in the investigation of differentiating minute concentrations of VOCs within intricate matrices [21]. HS-SPME-GC×GC-TOFMS is an advanced technique for detecting aromatic compounds, originally developed for the analysis of VOCs in dairy products [22], meat [23], liquor [24], tea [25], and other food and beverages. The technology offers notable benefits such as heightened sensitivity and effective separation, with a scanning rate that surpasses other mass spectrometry methods by 50–200 times. This ensures the acquisition of analytical information with high density and throughput, effectively addressing the issues of sluggish GC-MS analysis speed and the loss of VOCs resulting from pre-processing [26]. Currently, there is a lack of scholarly literature on the utilization of HS-SPME-GC×GC-TOFMS technology for the identification of VOCs in tomato fruits.

In light of the aforementioned issues, this study employed HS-SPME-GC-MS and HS-SPME-GC×GC-TOFMS techniques to comprehensively identify the VOCs present in cherry tomato fruits during the red ripening stage. Furthermore, a comparative analysis was conducted to assess the disparities in the detection outcomes obtained from both methodologies. Furthermore, the VOC composition of cherry tomato fruit was comprehensively examined by analyzing all detected VOCs using two different methods. The identification of key VOCs responsible for the aroma of cherry tomato was accomplished through the application of the rOAV method [27,28]. Additionally, the primary sensory flavor characteristics of cultivar var. ‘Zheyingfen1’ were elucidated. These findings offer valuable theoretical insights and practical guidance for the cultivation and production of superior cherry tomato cultivars. 

## 2. Materials and Methods

### 2.1. Test Material and Sampling Treatment

The cherry tomato cultivar var. ‘Zheyingfen1’ was newly bred by the Zhejiang Academy of Agricultural Sciences (ZAAS, Hangzhou, China). The experiments were carried out at the experimental farm of the ZAAS (longitude 120°2′ E, latitude 30°27′ N), Zhejiang Province, China, during the April to July 2023 growing season. The plants were spaced at a distance of 40 cm × 50 cm, and the fruits were harvested at the third and fourth stages of tomato commercial fruit ripening, characterized by more than 90% of the fruit exhibiting complete color transformation and optimal color and brightness. From a pool of more than five fruits, ten fruits of uniform size were randomly selected for analysis.

### 2.2. Sample Preparation and Treatment

The materials were collected, weighed, rapidly frozen in liquid nitrogen, and stored at −80 °C until required. Subsequently, the samples were pulverized into a fine powder using liquid nitrogen. A quantity of 500 mg (1 mL) of the powder was promptly transferred to a 20 mL head-space vial (Agilent, Palo Alto, CA, USA) containing a saturated solution of NaCl to prevent any enzymatic reactions. The vials were sealed using crimp-top caps equipped with TFE-silicone headspace septa (Agilent). During the SPME analysis, each vial was subjected to a temperature of 60 °C for 5 min, after which a 120 µm DVB/CWR/PDMS fiber (Agilent) was exposed to the headspace of the sample for a duration of 15 min.

### 2.3. Internal Standard Solution Preparation

A stock solution with 1 mg/L of n-Hexyl-d13 alcohol was prepared in 50% ethanol and stored in a 4 °C refrigerator. A stock solution with 1 mg/L of n-alkanes was prepared in n-hexane and stored in a 4 °C refrigerator.

### 2.4. HS-SPME-GC-MS Conditions

The powder sample, consisting of 500 mg (1 mL), was promptly transferred to a 20 mL head-space vial (Agilent, Palo Alto, CA, USA) containing a saturated NaCl solution to prevent any enzymatic reactions. The vials were sealed using crimp-top caps equipped with TFE-silicone headspace septa (Agilent). Prior to solid-phase microextraction (SPME) analysis, each vial was incubated at 60 °C for 5 min, followed by exposure of a 120 µm DVB/CWR/PDMS fiber (Agilent) to the headspace of the sample for 15 min at 60 °C.

After collecting samples, the VOCs were desorbed from the fiber coating in the injection port of the GC apparatus (Model 8890; Agilent) at a temperature of 250 °C for a duration of 5 min in the splitless mode. The identification and quantification of VOCs were conducted using an Agilent Model 8890 GC and a 7000D mass spectrometer (Agilent), which were equipped with a 30 m × 0.25 mm × 0.25 μm DB-5MS (5% phenyl-polymethylsiloxane) capillary column. Helium was utilized as the carrier gas with a linear velocity of 1.2 mL/min. The injector temperature was maintained at 250 °C, while the detector temperature was set at 280 °C. The oven temperature was programmed to increase from 40 °C [29].

### 2.5. HS-SPME-GC×GC TOFMS Conditions

The samples (0.5 g each) were placed in a headspace bottle and were extracted using SPME with a 1 cm DVB/CAR/PDMS fiber head and kept in a 60 °C bath for 40 min. The extracted samples were desorbed in the GC inlet for 5 min and subjected to GC×GCTOFMS analysis according to the set parameters.

The first-dimensional column utilized in the study was a DB-WAX column measuring 30 m in length, 250 μm in diameter, and 0.25 μm in film thickness. The injection temperature was set at 250 °C, with an initial temperature of 40 °C maintained for 3 min before being increased to 250 °C at a rate of 5 °C/min and held for 5 min. Helium gas with a purity of 99.9999% was used as the carrier gas at a flow rate of 1.0 mL/min with splitless injection. The second-dimensional column employed was a DB-17MS column measuring 2 m in length, 100 μm in diameter, and 0.10 μm in film thickness. The column temperature was consistently maintained at 5 °C higher than the first-dimensional column. The modulation period for full two-dimensional analysis was set at 6.0 s, with the interface temperature also maintained at a level 5 °C higher than the first-dimensional column [30,31,32]. 

### 2.6. Data Processing and Analysis

GC-MS qualitative and quantitative analysis: a database was created for GC-MS qualitative and quantitative analysis utilizing the NIST 2020 spectrum library as a foundation. This database incorporated various sources such as multiple species, literature, partial standards, and retention indexes. It included identified retention time (RT), as well as qualitative and quantitative ions for accurate scanning in selective ion detection mode. For each compound, one quantitative ion and 2–3 qualitative ions were chosen. In order to detect the qualitative ions in each group, it is necessary to detect them separately based on the peak order. The determination of the target material is made by comparing the detected retention time with the standard reference and observing the presence of the selected ions in the sample quality spectrum after background subtraction. To improve the accuracy of quantitative measurement, integration and correction of the selected quantitative ions are performed.

GC×GC-TOFMS qualitative and quantitative analysis: the VOCs’ mass spectrum was acquired using the LECO Pegasus BT 4D GC×GC-TOFMS instrument. The original data was annotated using the NIST 2020 database and Chroma TOF 4.3X database search software. This annotation process yielded information pertaining to the name, retention time, retention index, CAS number, and peak area of each substance. Subsequently, the relative concentrations of VOCs were standardized by normalizing them to the peak area of the internal standard material, deuterium-n-hexanol-D13. The calculation formula employed was IS: A/IS, where A represents the volatile organic compound under examination and IS denotes the peak area of the internal standard material within a given sample.

### 2.7. Calculating rOAV

In order to determine how each volatile component contributes to the overall flavor, the rOAV must be calculated. The formula for rOAV is as follows: rOAV = 100 × (Peak B*TA/Peak A*TB), where TA and Peak A were assumed to be the odor threshold and peak area of the compound with the minimum odor threshold in the sample, respectively. TB and Peak B are the odor threshold and peak area of the compound to be measured, respectively. Component A has a rOAV of 100 as a standard setting. Whenever rOAV ≥ 1, the compound is considered to be the key flavor compound.

### 2.8. Statistical Analysis

Data were expressed as the mean ± standard error of three replicates. Excel software (Excel 2016, Addinsoft, New York, NY, USA) and OriginPro 2022 (Origin Lab Corporation, Northampton, MA, USA) were used to collect data. Statistical analysis of the data was performed with IBM SPSS Statistics 26 (SPSS, Chicago, IL, USA). The data underwent testing to determine significant treatment differences (*p* < 0.05) using a one-way ANOVA, followed by Tukey’s test.

## 3. Results

### 3.1. Comparative Analysis of VOCs Components in Cherry Tomato by Different Techniques

After conducting a comparative analysis between the total ion flow chromatogram (Figure 1A) acquired through HS-SPME-GC-MS detection and the high-resolution mass spectrometry database, a total of 663 VOCs were successfully identified (Appendix A). The compound can be categorized into 15 distinct classes based on variations in chemical structure. These classes consist of terpenoids (120), esters (107), heterocyclics (104), hydrocarbons (103), ketones (58), aldehydes (56), alcohols (55), acids (18), aromatics (14), phenols (11), nitrogen compounds (5), halogenated hydrocarbons (4), sulfur compounds (3), ethers (3), and others (3) (Figure 1C). The quantity of terpenoids, esters, heterocyclics, and hydrocarbons detected exhibited a substantial increase compared to other substances, constituting over 15% individually and contributing to a total of 65.5% (Figure 1D). A comprehensive analysis was conducted using HS-SPME-GC×GC-TOFMS detection technology (Figure 1B), resulting in the identification and acquisition of a total of 1026 VOCs. These compounds encompass various classes, including hydrocarbons (305), heterocyclics (104), alcohols (99), ketones (96), esters (93), terpenoids (92), aldehydes (64), acids (41), nitrogen compounds (27), eths (16), halogenated hydrocarbons (6), phenols (5), sulfur compounds (3), alkaloids (1), and others (74). The utilization of this detection technique resulted in a notably elevated proportion of hydrocarbon components, surpassing that of other substances, with a magnitude of 29.7%. Additionally, the proportions of heterocyclic, terpenoid, ester, alcohol, aldehyde, and ketone substances were approximately 10% (Figure 1E). It is worth mentioning that this technique successfully identified one alkaloid that had remained undetected by the HS-SPME-GC-MS method (masonin).

Based on the analysis of VOC composition using two distinct methods, it was observed that 93.3% (14/15) of the identified types were consistent between the two approaches. Notably, only one of the methods, namely HS-SPME-GC-MS, successfully identified aromatics, while alkaloids were specifically detected using HS-SPME-GC×GC-TOFMS. Based on the data presented in Figure 1D,E, it is evident that the compounds detected through the HS-SPME-GC-MS assay constituted the majority of terpenoids, comprising 18.1% of the overall substances identified using this methodology. Conversely, the compounds identified through the HS-SPME-GC×GC-TOFMS assay accounted for a mere 9.0%. In the HS-SPME-GC×GC-TOFMS detection method, hydrocarbons constituted 29.7% of the overall substances identified, whereas the compounds identified solely by the HS-SPME-GC-MS method accounted for a mere 15.5%. Furthermore, esters, heterocyclics, ketones, aldehydes, alcohols, and five other compounds exhibited varying quantities in different detection methods, with their proportions being relatively similar.

The results of a quantitative comparison analysis revealed that the total number of VOCs detected using HS-SPME-GC×GC-TOFMS (1026) was 1.5 times greater than that detected using HS-SPME-GC-MS (663). Furthermore, the HS-SPME-GC-MS and HS-SPME-GC×GC-TOFMS techniques detected a total of 518 and 881 specific VOCs, respectively, with the latter exhibiting a 1.7-fold increase compared to the former (Figure 2A). Among the VOCs identified using the two techniques, a total of 145 VOCs were detected (Table 1). These included hydrocarbons (29), terpenes (23), aldehydes (21), ketones (19), alcohols (18), esters (18), heterocyclic compounds (6), phenols (5), acids (4), sulfur compound (1), and ether (1). Hydrocarbons constituted the largest proportion (20%) among the 145 VOCs, followed by terpenoids, aldehydes, ketones, alcohols, and esters. Collectively, these VOCs accounted for 88% of the total number (Figure 2B).

### 3.2. Comparative Analysis of VOC Content in Cherry Tomato

Based on a comparative analysis of qualitative findings, further examination was conducted on the relative quantitative results, revealing variations in the relative content of VOCs across different identification techniques. The HS-SPME-GC-MS identification yielded several results regarding the relative contents of volatile substances. Heterocyclics constituted the highest proportion at 27.1%, followed by terpenoids and hydrocarbons at 21.9% and 17.3%, respectively. Ketones, aldehydes, esters, alcohols, and phenols collectively accounted for a range of 2% to 10%. Nitrogen compounds, sulfur compounds, acids, aromatics, halogenated hydrocarbons, ethers, and others had relative contents of less than 1% (Figure 3A). In the HS-SPME-GC×GC-TOFMS detection method, alcohols comprised 21.8% of the detected compounds, while hydrocarbons, aldehydes, ketones, terpenes, and heterocyclic compounds accounted for 16.9%, 14.5%, 11.7%, 10.8%, and 10.6%, respectively. Each component exhibited a relative content exceeding 10%. Conversely, the relative content of VOCs belonging to five categories, namely ethers, sulfur compounds, phenols, halogenated hydrocarbons, and alkaloids, was less than 1% (Figure 3B).

Based on the analysis of various types of VOCs using two distinct methods, it was determined that heterocyclic compounds comprised the highest proportion of relative content (27.1%) in the HS-SPME-GC-MS detection method. Conversely, the HS-SPME-GC×GC-TOFMS method identified these substances with a significantly lower relative content of only 10.6%. The relative concentrations of alcohols and terpenes differed significantly between the two detection techniques, with values of 15.8% and 11.1%, respectively. Conversely, the relative concentrations of hydrocarbons, esters, and ketones were found to be similar across different detection methods, with differences of less than 3%.

### 3.3. Analysis of VOC Components in Cherry Tomato

In this study, a comprehensive analysis was conducted on a total of 1544 VOCs of cherry tomato using two different detection methods. The findings revealed the presence of 16 VOCs, which encompassed various chemical classes such as hydrocarbons (378), heterocyclics (201), terpenoids (189), esters (184), alcohols (136), ketones (135), aldehydes (99), others (77), acids (55), nitrogen-containing compounds (32), ethers (17), aromatics (14), phenols (11), halogenated hydrocarbons (10), sulfur compounds (5), and alkaloids (1). According to the data presented in Figure 4A, hydrocarbon VOCs exhibit the highest numerical count, constituting approximately 25% (Figure 4B), which is in close proximity to one-fourth of the overall VOC count. The quantities of heterocyclic compounds, terpenes, and esters exhibited a close proximity, collectively constituting 37% of the overall count. Conversely, ketones and alcohols accounted for approximately 9% of the total. The remaining constituents of VOCs are fewer than 100. The primary constituents of VOCs in cherry tomatoes include hydrocarbons, heterocyclics, terpenoids, esters, alcohols, and ketones, comprising a total of 1223 species. These components account for 79.2% of the overall substances present. Additionally, there are 321 species of aldehydes and others of compounds, acids, nitrogen compounds, ethers, aromatics, phenols, halogenated hydrocarbons, sulfur compounds, and alkaloids, which collectively represent only 21.8% of the total matter.

### 3.4. The Identification and Sensory Flavor Characterization of Crucial VOCs in Cherry Tomatoes

A total of 31 key VOCs were identified in cherry tomato fruits using a screening criterion of rOAV > 1 (Table 1). HS-SPME-GC-MS analysis revealed 25 VOCs with rOAV > 1, with aldehydes (7) being the most prevalent, followed by heterocyclics (6) and terpenoids, ketones, esters, alcohols, and other VOCs accounting for 12 compounds. Out of all the VOCs identified through screening, β-ionone, a terpenoid, exhibited the highest rOAV of 100.00. This compound is known for its sensory attributes resembling the fragrance of florals. Furthermore, the rOAV of 2-nonenal, (E)-(rOAV = 67.79), dimethyl trisulfur compounds (rOAV = 54.06), 2-thiophenemethanethiol (rOAV = 44.62), and pyrazine, 2-methoxy-3-(2-methylpropyl)- (rOAV = 39.60) exhibited a notable elevation compared to other VOCs. These aforementioned compounds predominantly contribute to the sensory attributes of tomato fruit, including the green, fatty, fresh, and coffee flavors. HS-SPME-GC×GC-TOFMS successfully identified a total of seven VOCs with rOAV > 1. Among these, four were classified as aldehydes, constituting 57.1% of the overall count. Additionally, two were identified as ketones, while one was categorized as heterocyclic. Based on the findings obtained from the method’s detection results, it was observed that the rOAV value (100.00) of 2-nonenal, (E)- exhibited a notably greater magnitude compared to the other constituents. Furthermore, this substance exhibited sensory attributes associated with green, fatty, and fresh. Among the remaining six substances, 2-Octenal, (E)- and furan, 2-pentyl- exhibited comparatively elevated rOAV values, primarily contributing to the sensory flavor attributes associated with green, fatty, and fragrant tomato fruits. Notably, the rOAV value of 2-nonenal, (E)- surpassed 50 in both techniques, underscoring its significant involvement in the composition of tomato aroma.

A flavor wheel was constructed through sensory descriptive analysis using two detection methods to identify 31 volatile substances with rOAV > 1. This flavor wheel effectively reflects the corresponding VOC categories and main flavors. Based on the analysis of VOC composition depicted in Figure 5, the predominant aroma attributes of cherry tomato fruits can be classified into 12 distinct categories, namely green, fatty, sweet, floral, fruity, floral, spicy, nut-like, coffee, terpene, hazelnut-like, and minty-like flavors. The findings indicated that during the ripening stage of cherry tomato fruit, the green flavor exhibited the highest frequency (13 times) among the VOCs. This was followed by the fatty flavor, which occurred 10 times, and the sweet flavor, which occurred 8 times. Additionally, the floral flavor and fruity flavor were observed 4 times and 3 times, respectively. The substance classifications and sensory characteristics of all VOCs listed in Table 1 have been addressed. The analysis of VOC composition in various flavors reveals the relatively intricate nature of VOC composition in cherry tomato fruit. Specifically, the three aroma sources, namely green flavor, sweet flavor, and floral flavor, exhibit diverse VOC compositions. The green flavor is characterized by the presence of ketones, heterocyclics, alcohols, and aldehydes. Conversely, the sweet flavor is associated with aldehydes, ketones, heterocyclics, and esters. Lastly, the floral aroma is derived from aldehydes, alcohols, esters, and terpenes. In contrast, the VOCs originating from the fatty flavor and fruity flavor sources exhibit a relatively straightforward composition. Specifically, the former predominantly consists of aldehydes, while the latter exclusively comprises ketones. The predominant factor contributing to the spicy flavor of tomato fruit is the presence of a single nitrogen compound, dodecanenitrile, alongside a hydrocarbon compound known as benzene, (2-nitroethyl)-. Additionally, it is noteworthy that nut-like, coffee, terpene, hazelnut-like, and minty-like flavors are characterized by the presence of a sole VOC.

To facilitate a comparison of the sensory flavor characteristics of cherry tomatoes obtained through two distinct detection methods, a sensory flavor characteristics radar map was constructed based on the cumulative contribution value of rOAV (Figure 6). Among the VOCs identified through the use of HS-SPME-GC-MS, it was observed that the oil fragrance exhibited the highest contribution value in terms of rOAV, with a cumulative contribution rOAV of 159.26. This was followed by the green fragrance, which had a cumulative contribution rOAV of 149.43, and the floral fragrance, which also had a cumulative contribution rOAV of 159.26. The VOCs identified through the use of HS-SPME-GC×GC-TOFMS revealed that the sensory attributes with the most significant contribution values were green (rOAV = 147.19), fatty (rOAV = 129.47), and fresh (rOAV = 100.00). These findings suggest that fatty and green flavors are prominent flavor profiles within the volatile composition of cherry tomatoes as detected in this particular study.

## 4. Discussion

The composition of the flavor of tomato fruit encompasses both taste and smell, with taste primarily influenced by sugars and organic acids and smell predominantly determined by VOCs [33]. VOCs are primarily characterized by distinct aromas and serve a significant function in enhancing the flavor profiles of agricultural products, specifically fruits and vegetables [34,35,36]. In recent years, the field of VOC detection has witnessed a surge in research activity due to the ongoing advancements in detection technology [37]. Previous research has indicated that investigations into the metabolism of VOCs in tomato fruits commenced as early as the 1950s [38,39]. However, limited advancements in detection technology, volatile substance composition databases, and analysis techniques have resulted in a relatively small number of identified VOCs in tomatoes compared to other fruit and vegetable crops. Consequently, this limitation hinders the comprehensive examination of the olfactory sensory characteristics and chemical constituents of tomatoes [40]. In order to address this issue, the present study focused on cherry tomato fruit as the subject of investigation and conducted a comprehensive analysis of its VOCs and aroma sensory flavor characteristics utilizing two volatile substance detection technologies, namely HS-SPME-GC-MS and HS-SPME-GC×GC-TOFMS.

The HS-SPME-GC-MS technique is currently regarded as a conventional means of detecting VOCs in fruit and vegetable crops. Through the utilization of this technology, diverse varieties of tomato fruits have been analyzed, leading to the identification of distinct types and quantities of VOCs. The results of Zhang et al. [41] show that a total of 110 VOCs (6 categories) were detected in the 15 tomato varieties. In addition, a total of 60 volatiles that were detected from the flavor of 71 tomato accessions have significant differences [42]. Based on the aforementioned test results, it is evident that the HS-SPME-GC-MS technique has a limited capacity to detect VOCs, typically not exceeding 200 in number and encompassing a limited range of types, typically no more than six distinct categories. Using the most recent VOC database and enhanced HS-SPME-GC-MS detection and analysis technology, a total of 663 VOCs belonging to 15 distinct categories, such as terpenoids, esters, and heterocyclics, were successfully identified in the cherry tomato cultivar var. ‘Zheyingfen1’. Throughout the postharvest period, the metabolic processes of climacteric fruits such as tomatoes result in significant alterations to the overall composition of the product [18]. The handling and storage practices during this stage play a crucial role in influencing fruit metabolism, ultimately affecting the components that contribute to flavor perception [43]. The storage temperature and duration significantly influence the evolution of aroma in postharvest fruit. The use of the liquid nitrogen quick-freezing method upon harvesting tomatoes effectively prevents alterations in VOCs during the preservation process, as demonstrated in this study. The present study reveals that both the type and quantity of VOCs are considerably higher compared to previous research findings, potentially attributable to the following three factors. The olfactory perception of cherry tomato fruit exhibited a notably superior quality compared to other tomato cultivars, accompanied by a greater complexity and diversity in its VOC constituents. Furthermore, the utilization of the HS-SPME-GC-MS detection technique in this study is characterized by a more advanced and superior detection instrument model compared to previous research. The identification of detected substances in this study was based on the utilization of an enhanced plant broad-target metabolism database, which encompasses a more extensive and comprehensive reference database. 

This study incorporated the latest HS-SPME-GC×GC-TOFMS technology alongside the HS-SPME-GC-MS method to detect VOCs in tomato fruit. The utilization of this method offers several advantages, including enhanced efficiency, speed, sensitivity, and resolution compared to HS-SPME-GC-MS. Consequently, it effectively broadens the scope of VOC detection [44]. The HS-SPME-GC×GC-TOFMS method exhibited a significantly higher detection of VOCs compared to the HS-SPME-GC-MS method, with an increase of approximately 55%. While both methods yielded similar results in terms of the number of substance categories detected (15 categories), notable differences were observed in category preference. Specifically, the HS-SPME-GC×GC-TOFMS method demonstrated greater sensitivity in detecting hydrocarbons, ketones, alcohols, and other substances, particularly hydrocarbons. Moreover, the number of identified species using this method exceeded that of HS-SPME-GC-MS by 202 [45]. It is noteworthy that the utilization of HS-SPME-GC×GC-TOFMS resulted in a significantly higher number of detected substances compared to HS-SPME-GC-MS. However, it is important to highlight that the combined number of substances detected by both techniques accounted for less than 10% of the total substances detected (145/1544). This finding contradicts the previous assumption that the substances identified by HS-SPME-GC-MS are encompassed within the substances detected by HS-SPME-GC×GC-TOFMS. This discrepancy may be attributed to the varying sensitivity of the two detection methods toward the identification of distinct substance types, as well as other potential factors. Hence, the detection of VOCs through a singular technology exhibits certain limitations. Employing multiple detection technology platforms, exceeding two, facilitates a more comprehensive and systematic analysis of aromatic flavor substances present in horticultural crops, such as tomatoes.

The perception of aroma by humans is achieved through the interaction between VOCs and receptors found in nasal epithelial cells [46,47]. Nevertheless, it is important to note that not all VOCs are associated with aroma perception by humans. Previous research has indicated that the impact on aroma is only observed when VOCs surpass a specific threshold [48]. The rOAV is commonly employed as a means of evaluating the impact of an individual VOC on the overall aroma. A higher rOAV indicates a more significant contribution of the corresponding VOC to the odor [49,50]. This study aimed to screen a total of 31 VOCs that impact the sensory aroma attributes of cherry tomatoes, using a standard of rOAV > 1. Additionally, the odor characteristics of cherry tomato fruit were comprehensively analyzed by constructing a flavor wheel and a sensory flavor characteristics radar map. Research has demonstrated that aldehydes significantly contribute to the aromatic composition of tomato fruits. Aldehydes are typically produced via the lipoxygenase (LOX) pathway, which involves the oxidation, cleavage, and reduction of fatty acids and other precursor compounds. Linoleic acid and alpha-linolenic acid, the primary precursors, were initially obtained from acylglycerol. These fatty acids undergo lipid degradation through the enzymatic actions of lipoxygenase (TomLoxC) and 13-hydroperoxide lyase (13-HPL), resulting in the generation of a range of short-chain C6 and C5 aldehydes and their respective alcohols. Consequently, genes associated with pivotal enzymes in the fatty acid pathway may exert a crucial influence on the regulation and modulation of tomato aroma constituents [51]. 

Furthermore, previous research has substantiated the presence of two volatile secondary metabolites, namely β-ionone and (E)-2-nonenal, which significantly contribute to the development of the distinctive flavor profile observed in cherry tomato fruits. β-ionone primarily contributes to the aromatic properties of tomato fruits, exhibiting a positive correlation with the desirability of tomato flavor and exerting a favorable influence on the enhancement of tomato flavor. This finding aligns with the empirical findings of Martina [52]. The sensory properties of (E)-2-nonenal in fruit include green, fatty, and fresh, which are characterized by pleasant and fragrant attributes. This aligns with the findings of Liu [53], who determined that (E)-2-nonenal plays a significant role in fruit aroma. The compounds dimethyl trisulfur, 2-octenal, (E)-, and furan, 2-pentyl-, which were identified in this study, have also been the subject of previous investigation [42,54,55]. It is noteworthy that this study has also identified two substances, namely 2-thiophenemethanethiol and pyrazine, 2-methoxy-3-(2-methylpropyl)-, which significantly contribute to the olfactory perception of tomato fruits. These substances are known to impart coffee and green flavors to tomatoes and have been established as the primary VOCs in sesame oil and black tea [56,57]. Currently, there is limited literature documenting the effects of these two substances on various horticultural crops, including tomatoes. Additionally, the present study identified the presence of a single alkaloid compound. According to Kazachkova et al. [58], this alkaloid, specifically α-tomatidine, has been linked to the development of a bitter taste in fruits. 

The composition of tomato fruit odor is influenced by various types of substances, each exerting distinct effects, with certain substances playing significant roles in shaping the overall composition of tomato fruit odor. In contrast to measurable breeding traits such as yield, disease resistance, shape, and color, the constituents of tomato flavor are comprised of various volatile substances. According to the varying contribution rates of key VOCs associated with the rOAV identified in this investigation, it has been determined that aldehydes possess the potential to enhance the verdant aroma of fruits, while terpenoids are capable of imparting a floral scent to fruits. The predominant sensory flavor attributes exhibited by cherry tomato fruits primarily encompass a green and fatty essence, accompanied by both fragrant and floral olfactory perceptions. Simultaneously, it is postulated that the enzymes associated with the fatty acid pathway exert an influence on the expression levels of distinct genes implicated in this metabolic pathway, thus offering a theoretical rationale for the development of the cherry tomato flavor.

## 5. Conclusions

The investigation of VOCs in tomato fruit holds substantial scientific importance in enhancing flavor quality. This research focuses on cherry tomatoes as the subject of study, and it compares and evaluates the efficacy of HS-SPME-GC-MS and HS-SPME-GC×GC-TOFMS in detecting tomato VOCs. Through the analysis of the detection outcomes, a comprehensive and systematic examination of the sensory aroma attributes and associated VOCs in tomato fruits is conducted. Despite comparing and integrating two techniques for detecting volatile compounds, this study still has limitations. In subsequent research endeavors, the prominent VOCs identified in this investigation will be subjected to further examination and employed in conjunction with more pertinent methodologies, such as gas chromatography–ion mobility spectroscopy (GC-IMS), the gas chromatography olfactory method (GC-O), electronic nose, and so on. The anticipated outcomes of this study are poised to establish a fundamental basis for the exploration of odor characteristics analysis, VOC composition, and metabolic regulation in horticultural crops, particularly tomatoes. Additionally, this research is expected to furnish theoretical guidance and technical assistance for the cultivation of superior tomato cultivars based on fragrance attributes, as well as the development of corresponding cultivation techniques.

## Figures and Tables

**Figure 1 foods-13-01279-f001:**
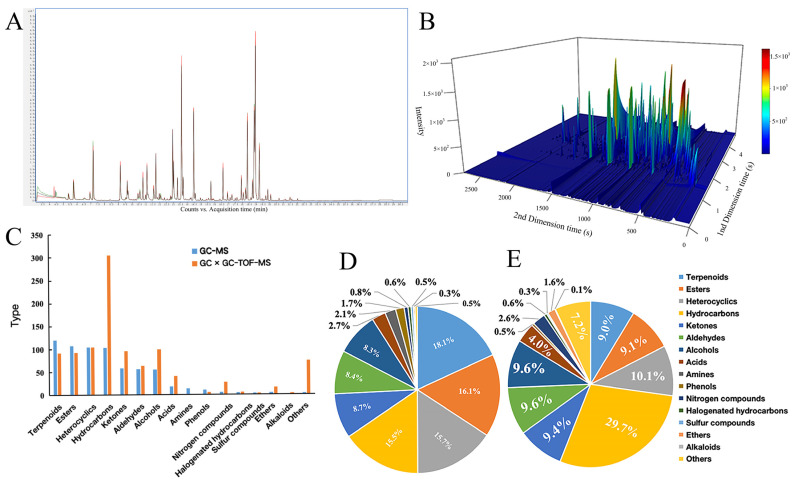
Effect of different detection methods on VOCs in cherry tomato. (**A**) Total ion flow diagram of VOCs detected by HS-SPME-GC-MS; (**B**) 3D chromatogram obtained by HS-SPME-GC×GC-TOFMS detection; (**C**) variation of VOCs in two identification methods; (**D**) HS-SPME-GC-MS detected the proportion of VOCs of each component; (**E**) HS-SPME-GC×GC-TOFMS detected the proportion of VOCs of each component.

**Figure 2 foods-13-01279-f002:**
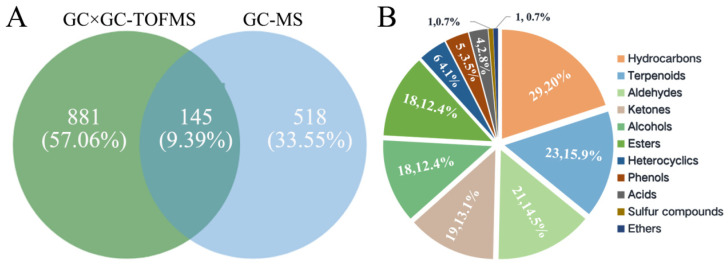
Analysis of VOCs in cherry tomato by different detection methods. (**A**) Venn diagrams of VOCs identified by two detection methods; (**B**) the quantity ratio of each component of common VOCs identified by the two detection methods.

**Figure 3 foods-13-01279-f003:**
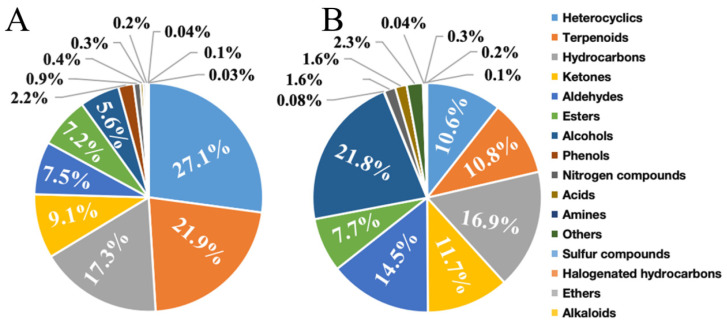
Detection effects of different detection methods on relative VOCs in cherry tomato. (**A**) The relative content of different categories of VOCs obtained via HS-SPME-GC-MS identification; (**B**) relative contents of VOCs of different categories identified via HS-SPME-GC×GC-TOFMS.

**Figure 4 foods-13-01279-f004:**
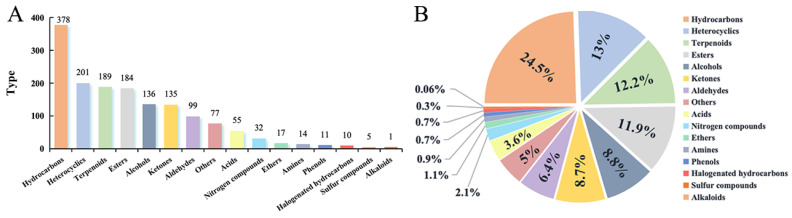
Detection effects of different detection methods on relative VOCs in cherry tomato. (**A**) The relative content of different categories of VOCs obtained by HS-SPME-GC-MS identification; (**B**) relative contents of VOCs of different categories identified by HS-SPME-GC×GC-TOFMS.

**Figure 5 foods-13-01279-f005:**
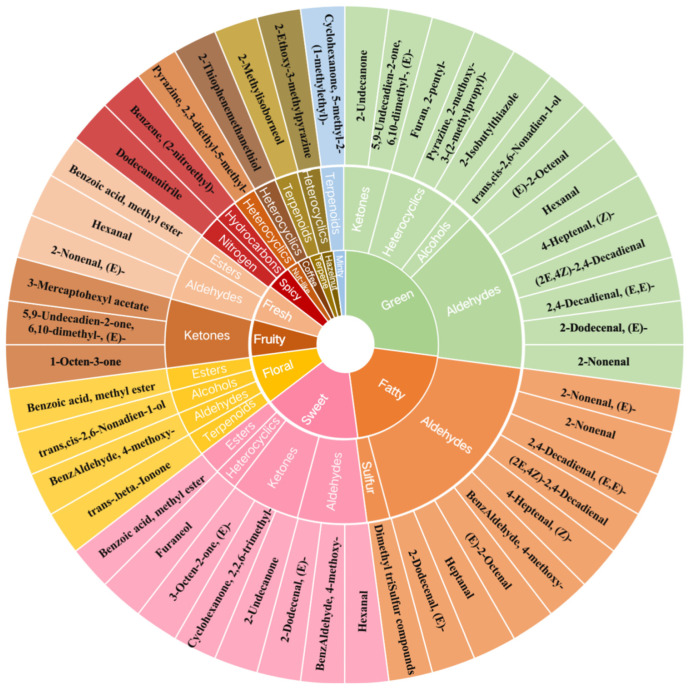
The flavor wheel of key VOCs in cherry tomatoes. Note: the outermost ring represents the name of the VOC, the middle ring represents the corresponding classification of the VOC, and the innermost circle represents the sensory flavor characteristics of VOCs.

**Figure 6 foods-13-01279-f006:**
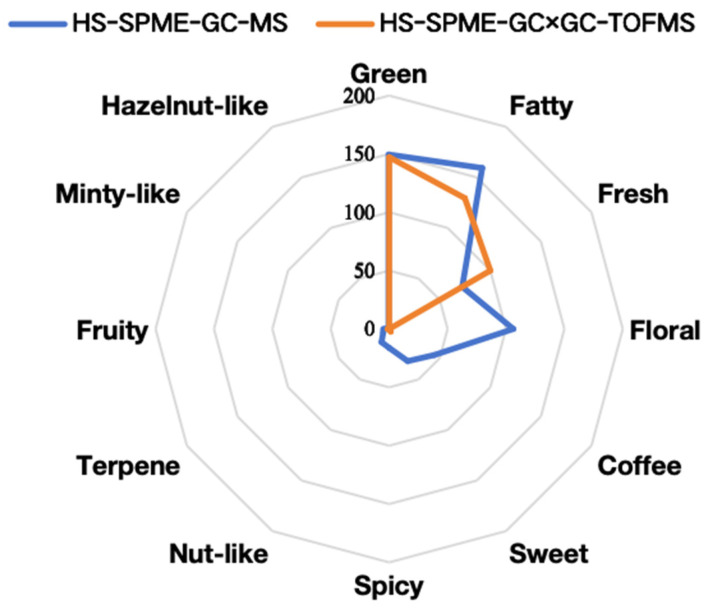
Radar map of aroma sensory characteristics of key VOCs in cherry tomato. Note: the outermost ring indicates sensory flavor characteristics, and the broken line represents the cumulative contribution rate of rOAV of the corresponding type.

**Table 1 foods-13-01279-t001:** 31 VOCs (rOAV > 1) in cherry tomato detected by HS-SPME-GC-MS and HS-SPME-GC×GC-TOFMS.

Name	Type	Sensory Flavor Characteristics	rOAV
HS-SPME-GC-MS			
β-ionone	Terpenoids	Floral	100.00
2-Nonenal, (E)-	Aldehydes	Green/fatty/fresh	67.79
Dimethyl triSulfur compounds	Sulfur compounds	Fatty	54.06
2-Thiophenemethanethiol	Heterocyclics	Coffee	44.62
Pyrazine, 2-methoxy-3-(2-methylpropyl)-	Heterocyclics	Green	39.60
2,4-Decadienal, (E,E)-	Aldehydes	Green/fatty	16.58
Furaneol	Heterocyclics	Sweet	13.84
Pyrazine, 2,3-diethyl-5-methyl-	Heterocyclics	Nut-like	13.04
Dodecanenitrile	Nitrogen compounds	Spicy	11.92
(2E,4Z)-2,4-Decadienal	Aldehydes	Green/fatty	8.57
2-Methylisoborneol	Terpenoids	Terpene	5.90
3-Octen-2-one	Ketones	Sweet	4.91
4-Heptenal, (Z)-	Aldehydes	Green/fatty	4.85
2-Nonenal	Aldehydes	Green/fatty	4.67
Benzene, (2-nitroethyl)-	Hydrocarbons	Spicy	4.29
Hexanal	Aldehydes	Green/sweet/fresh	3.13
3-Mercaptohexyl acetate	Esters	Fruity	2.91
BenzAldehyde, 4-methoxy-	Aldehydes	Fatty/sweet/floral	2.74
trans,cis-2,6-Nonadien-1-ol	Alcohols	Green/floral	2.22
2-Isobutylthiazole	Heterocyclics	Green	2.02
Cyclohexanone, 5-methyl-2-(1-methylethyl)-	Terpenoids	Minty-like	1.90
Benzoic acid, methyl ester	Esters	Sweet/floral/fresh	1.29
2-Ethoxy-3-methylpyrazine	Heterocyclics	Hazelnut-like	1.14
5,9-Undecadien-2-one, 6,10-dimethyl-, (E)-	Ketones	Fruity	1.13
HS-SPME-GC×GC-TOF-MS			
2-Nonenal, (E)-	Aldehydes	Green/fatty/fresh	100
2-Octenal, (E)-	Aldehydes	Green/fatty	24.52
Furan, 2-pentyl-	Heterocyclics	Green	20.02
Heptanal	Aldehydes	Fatty	3.31
2-Dodecenal, (E)-	Aldehydes	Green/fatty/sweet	1.64
1-Octen-3-one	Ketones	Fruity	1.30
2-Undecanone	Ketones	Green/sweet	1.01

Material level category mainly comes from https://pubchem.ncbi.nlm.nih.gov (accessed on 1 January 2024). Odor mainly comes from http://www.thegoodscentscompany.com (accessed on 1 January 2024).

## Data Availability

The original contributions presented in the study are included in the article/Appendix A further inquiries can be directed to the corresponding author.

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
