# Peer review of "Detection and Analysis of VOCs in Cherry Tomato Based on GC-MS and GC×GC-TOF MS Techniques"

_foods, 2024, doi:10.3390/foods13081279_

Round 1

Reviewer 1 Report

Comments and Suggestions for Authors

Thank you very much for your interesting reserach. Some points must be carefully revised:

INTRODUCTION. Line 36. It seems to be a typo in “significanc”

INTRODUCTION. Please, revise the use of italics carefully throughout the manuscript, for instance, in “Solanum”.

INTRODUCTION. What about those VOCs that are linked to off-flavors? Or those indicating rot, infection, overripeness, etc.?

MATERIALS AND METHODS. Lines 112-119. Do you think this process for tomato preservation might affect the VOCs profile? Could you please include a statement related to in the Discussion?

MATERIALS AND METHODS. Statistical analysis section is required.

RESULTS. Figure 4. The color codes that were chosen for Figure 4A and Figure 4B are different. Please, revise it carefully to avoid confusion to the readers.

DISCUSSION. Line 377. ‘Products’ instead of ‘produce’?

DISCUSSION. In my opinion, perhaps the discussion should be divided in different subsections, following the structure of the presentation of the obtained results.

CONCLUSIONS. Please, include current limitations and future perspectives.

Reviewer 2 Report

Comments and Suggestions for Authors

The study is well structured and robust in its analysis of results. However, some revisions, major and minor, are needed. In fact, some essential elements are missing in the analytical method part.

Minor revisions:

Line 97: detail the rOAV method and support it with references.

Line 108: What criteria are used to evaluate fruit transformation and optimal color? How is the selection carried out?

Line 212: how were the HS-SPME parameters chosen? 

Line 123: specifies the composition of the ISTD solution.

Line 162: "A" in lower case. 

The rOAV method must be described in the materials and methods. 

The Figure 1 is too small and difficult to read. Please consider editing it or creating more figures. 

Major revisions:

Essential information about the limits of detection (LOD) and limits of quantification (LOQ) of the analytical method are missing. Please provide these information in a general way (based on a STD that the authors consider to be representative) or at least for one significant compound from each of the three most significant compound classes (type). There is no information on VOC loss during sample preparation and treatment. Therefore, information regarding the recovery rate is also missing. Please provide the recovery rate for 2-3 of the compounds considered most significant (or an STD that the authors consider as representative) by doping the samples before preparation (injection into the fruit or addition during the pulverization step).

Round 2

Reviewer 1 Report

Comments and Suggestions for Authors

Thank you for addressing all the comments in the revision process.

Reviewer 2 Report

Comments and Suggestions for Authors

The changes meet my requirements, thank you